# An Integer Polynomial Programming Based Framework for Lifted MAP Inference

**Somdeb Sarkhel, Deepak Venugopal**
Computer Science Department
The University of Texas at Dallas
{sxs104721,dxv021000}@utdallas.edu

**Parag Singla**
Department of CSE
I.I.T. Delhi
parags@cse.iitd.ac.in

**Vibhav Gogate**
Computer Science Department
The University of Texas at Dallas
vgogate@hlt.utdallas.edu

## Abstract

In this paper, we present a new approach for lifted MAP inference in Markov logic networks (MLNs). The key idea in our approach is to compactly encode the MAP inference problem as an Integer Polynomial Program (IPP) by schematically applying three lifted inference steps to the MLN: lifted decomposition, lifted conditioning, and partial grounding. Our IPP encoding is lifted in the sense that an integer assignment to a variable in the IPP may represent a truth-assignment to multiple indistinguishable ground atoms in the MLN. We show how to solve the IPP by first converting it to an Integer Linear Program (ILP) and then solving the latter using state-of-the-art ILP techniques. Experiments on several benchmark MLNs show that our new algorithm is substantially superior to ground inference and existing methods in terms of computational efficiency and solution quality.

## 1 Introduction

Many domains in AI and machine learning (e.g., NLP, vision, etc.) are characterized by rich relational structure as well as uncertainty. Statistical relational learning (SRL) models [5] combine the power of first-order logic with probabilistic graphical models to effectively handle both of these aspects. Among a number of SRL representations that have been proposed to date, Markov logic [4] is arguably the most popular one because of its simplicity; it compactly represents domain knowledge using a set of weighted first order formulas and thus only minimally modifies first-order logic.

The key task over Markov logic networks (MLNs) is inference which is the means of answering queries posed over the MLN. Although, one can reduce the problem of inference in MLNs to inference in graphical models by propositionalizing or grounding the MLN (which yields a Markov network), this approach is not scalable. The reason is that the resulting Markov network can be quite large, having millions of variables and features. One approach to achieve scalability is *lifted inference*, which operates on groups of indistinguishable random variables rather than on individual variables. Lifted inference algorithms identify groups of indistinguishable atoms by looking for symmetries in the first-order logic representation, grounding the MLN only as necessary. Naturally, when the number of such groups is small, lifted inference is significantly better than propositional inference.

Starting with the work of Poole [17], researchers have invented a number of lifted inference algorithms. At a high level, these algorithms "lift" existing probabilistic inference algorithms (cf. [3, 6, 7, 21, 22, 23, 24]). However, many of these lifted inference algorithms have focused on the task of marginal inference, i.e., finding the marginal probability of a ground atom given evidence. For many problems

of interest such as in vision and NLP, one is often interested in the MAP inference task, i.e., finding the most likely assignment to all ground atoms given evidence. In recent years, there has been a growing interest in lifted MAP inference. Notable lifted MAP approaches include exploiting uniform assignments for lifted MPE [1], lifted variational inference using graph automorphism [2], lifted likelihood-maximization for MAP [8], exploiting symmetry for MAP inference [15] and efficient lifting of MAP LP relaxations using k-locality [13]. However, a key problem with most of the existing lifted approaches is that they require significant modifications to be made to propositional inference algorithms, and for optimal performance require lifting several steps of propositional algorithms. This is time consuming because one has to lift decades of advances in propositional inference.

To circumvent this problem, recently Sarkhel et al. [18] advocated using the "lifting as pre-processing" paradigm [20]. The key idea is to apply lifted inference as pre-processing step and construct a Markov network that is lifted in the sense that its size can be much smaller than ground Markov network and a complete assignment to its variables may represent several complete assignments in the ground Markov network. Unfortunately, Sarkhel et al.'s approach does not use existing research on lifted inference to the fullest extent and is efficient only when first-order formulas have no shared terms.

In this paper, we propose a novel lifted MAP inference approach which is also based on the "lifting as pre-processing" paradigm but unlike Sarkhel et al.'s approach is at least as powerful as probabilistic theorem proving [6], an advanced lifted inference algorithm. Moreover, our new approach can easily subsume Sarkhel et al.'s approach by using it as just another lifted inference rule. The key idea in our approach is to reduce the lifted MAP inference (maximization) problem to an equivalent *Integer Polynomial Program* (IPP). Each variable in the IPP potentially refers to an assignment to a large number of ground atoms in the original MLN. Hence, the size of search space of the generated IPP can be significantly smaller than the ground Markov network.

Our algorithm to generate the IPP is based on the following three lifted inference operations which incrementally build the polynomial objective function and its associated constraints: (1) Lifted decomposition [6] finds sub-problems with identical structure and solves only one of them; (2) Lifted conditioning [6] replaces an atom with only one logical variable (singleton atom) by a variable in the integer polynomial program such that each of its values denotes the number of the true ground atoms of the singleton atom in a solution; and (3) Partial grounding is used to simplify the MLN further so that one of the above two operations can be applied.

To solve the IPP generated from the MLN we convert it to an equivalent zero-one Integer Linear Program (ILP) using a classic conversion method outlined in [25]. A desirable characteristic of our reduction is that we can use any off-the-shelf ILP solver to get exact or approximate solution to the original problem. We used a parallel ILP solver, Gurobi [9] for this purpose. We evaluated our approach on multiple benchmark MLNs and compared with Alchemy [11] and Tuffy [14], two state-of-the-art MLN systems that perform MAP inference by grounding the MLN, as well as with the lifted MAP inference approach of Sarkhel et al. [18]. Experimental results show that our approach is superior to Alchemy, Tuffy and Sarkhel et al.'s approach in terms of scalability and accuracy.

## 2   Notation And Background

**Propositional Logic.** In propositional logic, sentences or formulas, denoted by $f$, are composed of symbols called propositions or atoms, denoted by upper case letters (e.g., $X$, $Y$, $Z$, etc.) that are joined by five logical operators $\wedge$ (conjunction), $\vee$ (disjunction), $\neg$ (negation), $\Rightarrow$ (implication) and $\Leftrightarrow$ (equivalence). Each atom takes values from the binary domain $\{true, false\}$.

**First-order Logic.** An atom in first-order logic (FOL) is a predicate that represents relations between objects. A predicate consists of a predicate symbol, denoted by Monospace fonts (e.g., `Friends`, `R`, etc.), followed by a parenthesized list of arguments. A *term* is a logical variable, denoted by lower case letters such as $x$, $y$, and $z$, or a constant, denoted by upper case letters such as $X$, $Y$, and $Z$. We assume that each logical variable, say $x$, is typed and takes values from a finite set of constants, called its *domain*, denoted by $\Delta x$. In addition to the logical operators, FOL includes universal $\forall$ and existential $\exists$ quantifiers. Quantifiers express properties of an entire collection of objects. A formula in first order logic is an atom, or any complex sentence that can be constructed from atoms using logical operators and quantifiers. For example, the formula $\forall x$ `Smokes`$(x) \Rightarrow$ `Asthma`$(x)$ states that all persons who smoke have asthma. A *Knowledge base* (KB) is a set of first-order formulas.

In this paper we use a subset of FOL which has no function symbols, equality constraints or existential quantifiers. We assume that formulas are *standardized apart*, namely no two formulas share a logical variable. We also assume that domains are finite and there is a one-to-one mapping between constants and objects in the domain (Herbrand interpretations). We assume that each formula $f$ is of the form $\forall \mathbf{x} f$, where $\mathbf{x}$ is the set of variables in $f$ (also denoted by $V(f)$) and $f$ is a disjunction of literals (clause); each literal being an atom or its negation. For brevity, we will drop $\forall$ from all formulas. A *ground atom* is an atom containing only constants. A *ground formula* is a formula obtained by substituting all of its variables with a constant, namely a formula containing only ground atoms. A *ground KB* is a KB containing all possible groundings of all of its formulas.

**Markov Logic.** Markov logic [4] extends FOL by softening hard constraints expressed by formulas and is arguably the most popular modeling language for SRL. A soft formula or a weighted formula is a pair $(f, w)$ where $f$ is a formula in FOL and $w$ is a real-number. A Markov logic network (MLN), denoted by $M$, is a set of weighted formulas $(f_i, w_i)$. Given a set of constants that represent objects in the domain, a Markov logic network represents a Markov network or a log-linear model. The ground Markov network is obtained by grounding the weighted first-order knowledge base with one feature for each grounding of each formula. The weight of the feature is the weight attached to the formula. The ground network represents the probability distribution $P(\omega) = \frac{1}{Z} \exp\left(\sum_i w_i N(f_i, \omega)\right)$ where $\omega$ is a world, namely a truth-assignment to all ground atoms, $N(f_i, \omega)$ is the number of groundings of $f_i$ that evaluate to $true$ given $\omega$ and $Z$ is a normalization constant.

For simplicity, we will assume that the MLN is in normal form and has no self joins, namely no two atoms in a formula have the same predicate symbol [10]. A *normal* MLN is an MLN that satisfies the following two properties: (i) there are no constants in any formula; and (ii) If two distinct atoms of predicate R have variables $x$ and $y$ as the same argument of R, then $\Delta x = \Delta y$. Because of the second condition, in normal MLNs, we can associate domains with each argument of a predicate. Moreover, for inference purposes, in normal MLNs, we do not have to keep track of the actual elements in the domain of a variable, all we need to know is the size of the domain [10]. Let $i_{\texttt{R}}$ denote the $i$-th argument of predicate R and let $D(i_{\texttt{R}})$ denote the number of elements in the domain of $i_{\texttt{R}}$. Henceforth, we will abuse notation and refer to normal MLNs as MLNs.

**MAP Inference in MLNs.** A common optimization inference task over MLNs is finding the most probable state of the world $\omega$, that is finding a complete assignment to all ground atoms which maximizes the probability. Formally,

$$\arg\max_{\omega} P_{\mathcal{M}}(\omega) = \arg\max_{\omega} \frac{1}{Z(\mathcal{M})} \exp\left(\sum_i w_i N(f_i, \omega)\right) = \arg\max_{\omega} \sum_i w_i N(f_i, \omega) \qquad (1)$$

From Eq. (1), we can see that the MAP problem in Markov logic reduces to finding the truth assignment that maximizes the sum of weights of satisfied clauses. Therefore, any weighted satisfiability solver can used to solve this problem. The problem is NP-hard in general, but effective solvers exist, both exact and approximate. Examples of such solvers are MaxWalkSAT [19], a local search solver and Clone [16], a branch-and-bound solver. Both these algorithms are propositional and therefore they are unable to exploit relational structure that is inherent to MLNs.

**Integer Polynomial Programming (IPP).** An IPP problem is defined as follows:

$$\begin{aligned} &\text{Maximize} && f(x_1, x_2, ..., x_n) \\ &\text{Subject to} && g_j(x_1, x_2, ..., x_n) \geq 0 \quad (j = 1, 2, ..., m) \end{aligned}$$

where each $x_i$ takes finite integer values, and the objective function $f(x_1, x_2, ..., x_n)$, and each of the constraints $g_j(x_1, x_2, ..., x_n)$ are polynomials on $x_1, x_2, ..., x_n$. We will compactly represent an integer polynomial programming problem (IPP) as an ordered triple $\mathcal{I} = \langle f, G, X \rangle$, where $X = \{x_1, x_2, ..., x_n\}$, and $G = \{g_1, g_2, ..., g_m\}$.

## 3   Probabilistic Theorem Proving Based MAP Inference Algorithm

We motivate our approach by presenting in Algorithm 1, the most basic algorithm for lifted MAP inference. Algorithm 1 extends the probabilistic theorem proving (PTP) algorithm of Gogate and Domingos [6] to MAP inference and integrates it with Sarkhel et al's lifted MAP inference rule [18]. It is obtained by replacing the summation operator in the conditioning step of PTP by the maximization operator (PTP computes the partition function). Note that throughout the paper, we will present

algorithms that compute the MAP value rather than the MAP assignment; the assignment can be recovered by tracing back the path that yielded the MAP value. We describe the steps in Algorithm 1 next, starting with some required definitions.

Two arguments $i_{\mathtt{R}}$ and $j_{\mathtt{S}}$ are called unifiable if they share a logical variable in a MLN formula. Clearly, unifiable, if we consider it as a binary relation $U(i_{\mathtt{R}}, j_{\mathtt{S}})$ is symmetric and reflexive. Let $\mathcal{U}$ be the transitive closure of $U$. Given an argument $i_{\mathtt{S}}$, let $\mathrm{Unify}(i_{\mathtt{S}})$ denote the equivalence class under $\mathcal{U}$.

**Simplification.** In the simplification step, we simplify the predicates possibly reducing their arity (cf. [6, 10] for details). An example simplification step is the following: if no atoms of a predicate share logical variables with other atoms in the MLN then we can replace the predicate by a new predicate having just one argument; the domain size of the argument is the product of domain sizes of the individual arguments.

---

**Algorithm 1 PTP-MAP**(MLN $M$)

  **if** $M$ is empty **return** $0$
  Simplify($M$)
  **if** $M$ has *disjoint* MLNs $M_1, \ldots, M_k$ **then**
     **return** $\sum_{i=1}^{k}$ PTP-MAP($M_i$)
  **if** $M$ has a *decomposer* $\mathbf{d}$ such that $D(i \in \mathbf{d}) > 1$ **then**
     **return** PTP-MAP($M|\mathbf{d}$)
  **if** $M$ has an *isolated atom* $\mathtt{R}$ such that $D(i_{\mathtt{R}}) > 1$ **then**
     **return** PTP-MAP ($M|\{1_{\mathtt{R}}\}$)
  **if** $M$ has a *singleton atom* $\mathtt{A}$ **then**
     **return** $\max_{i=0}^{D(1_{\mathtt{A}})}$ PTP-MAP($M|(\mathtt{A}, i)$) $+ w(\mathtt{A}, i)$
  Heuristically select an argument $i_{\mathtt{R}}$
  **return** PTP-MAP($M|G(i_{\mathtt{R}})$)

---

**Example 1.** *Consider a normal MLN with two weighted formulas:* $\mathtt{R}(x_1, y_1) \vee \mathtt{S}(z_1, u_1), w_1$ *and* $\mathtt{R}(x_2, y_2) \vee \mathtt{S}(z_2, u_2) \vee \mathtt{T}(z_2, v_2), w_2$. *We can simplify this MLN by replacing* $\mathtt{R}$ *by a predicate* $\mathtt{J}$ *having one argument such that* $D(1_{\mathtt{J}}) = D(1_{\mathtt{R}}) \times D(2_{\mathtt{R}})$. *The new MLN has two formulas:* $\mathtt{J}(x_1) \vee \mathtt{S}(z_1, u_1), w_1$ *and* $\mathtt{J}(x_2) \vee \mathtt{S}(z_2, u_2) \vee \mathtt{T}(z_2, v_2), w_2$.

**Decomposition.** If an MLN can be decomposed into two or more disjoint MLNs sharing no first-order atom, then the MAP solution is just a sum over the MAP solutions of all the disjoint MLNs.

**Lifted Decomposition.** Main idea in lifted decomposition [6] is to identify identical but disconnected components in ground Markov network by looking for symmetries in the first-order representation. Since the disconnected components are identical, only one of them needs to be solved and the MAP value is the MAP value of one of the components times the number of components. One way of identifying identical disconnected components is by using a decomposer [6, 10], defined below.

**Definition 1.** *[Decomposer] Given a MLN $M$ having $m$ formulas denoted by $f_1, \ldots, f_m$, $\mathbf{d} = \mathrm{Unify}(i_{\mathtt{R}})$ where $\mathtt{R}$ is a predicate in $M$, is called a decomposer iff the following conditions are satisfied: (i) for each predicate $\mathtt{R}$ in $M$ there is exactly one argument $i_{\mathtt{R}}$ such that $i_{\mathtt{R}} \in \mathbf{d}$; and (ii) in each formula $f_i$, there exists a variable $x$ such that $x$ appears in all atoms of $f_i$ and for each atom having predicate symbol $\mathtt{R}$ in $f_i$, $x$ appears at position $i_{\mathtt{R}} \in \mathbf{d}$.*

Denoted by $M|\mathbf{d}$ the MLN obtained from $M$ by setting domain size of all elements $i_{\mathtt{R}}$ of $\mathbf{d}$ to one and updating weight of each formula that mentions $\mathtt{R}$ by multiplying it by $D(i_{\mathtt{R}})$. We can prove that:

**Proposition 1.** *Given a decomposer $\mathbf{d}$, the MAP value of $M$ is equal to the MAP value of $M|\mathbf{d}$.*

**Example 2.** *Consider a normal MLN $M$ having two weighted formulas $\mathtt{R}(x) \vee \mathtt{S}(x), w_1$ and $\mathtt{R}(y) \vee \mathtt{T}(y), w_2$ where $D(1_{\mathtt{R}}) = D(1_{\mathtt{S}}) = D(1_{\mathtt{T}}) = n$. Here, $\mathbf{d} = \{1_{\mathtt{R}}, 1_{\mathtt{S}}, 1_{\mathtt{T}}\}$ is a decomposer. The MLN $M|\mathbf{d}$ is the MLN having the same two formulas as $M$ with weights updated to $nw_1$ and $nw_2$ respectively. Moreover, in the new MLN $D(1_{\mathtt{R}}) = D(1_{\mathtt{S}}) = D(1_{\mathtt{T}}) = 1$.*

**Isolated Singleton Rule.** Sarkhel et al. [18] proved that if the MLN $M$ has an isolated predicate $\mathtt{R}$ such that all atoms of $\mathtt{R}$ do not share any logical variables with other atoms, then one of the MAP solutions of $M$ has either all ground atoms of $\mathtt{R}$ set to $true$ or all of them set to $false$, namely, the solution lies at the extreme assignments to groundings of $\mathtt{R}$. Since we simplify the MLN, all such predicates $\mathtt{R}$ have only one argument, namely, they are singleton. Therefore, the following proposition is immediate:

**Proposition 2.** *If $M$ has an isolated singleton predicate $\mathtt{R}$, then the MAP value of $M$ equals the MAP value of $M|\{1_{\mathtt{R}}\}$ (the notation $M|\{1_{\mathtt{R}}\}$ is defined just after the definition of the decomposer).*

**Lifted Conditioning over Singletons.** Performing a conditioning operation on a predicate means conditioning on all possible ground atoms of that predicate. Naïvely it can result in exponential

number of alternate MLNs that need to be solved, one for each assignment to all groundings of the predicate. However if the predicate is singleton, we can group these assignments into equi-probable sets based on number of true groundings of the predicate (*counting assignment*) [6, 10, 12]. In this case, we say that the lifted conditioning operator is applicable. For a singleton $A$, we denote the counting assignment as the ordered pair $(A, i)$ which the reader should interpret as exactly $i$ groundings of $A$ are $true$ and the remaining are $false$.

We denote by $M|(A, i)$ the MLN obtained from $M$ as follows. For each element $j_R$ in $\text{Unify}(1_A)$ (in some order), we split the predicate $R$ into two predicates $R_1$ and $R_2$ such that $D(j_{R_1}) = i$ and $D(j_{R_2}) = D(1_A) - i$. We then rewrite all formulas using these new predicate symbols. Assume that $A$ is split into two predicates $A_1$ and $A_2$ respectively with $D(1_{A_1}) = i$ and $D(1_{A_2}) = D(1_A) - i$. Then, we delete all formulas in which either $A_1$ appears positively or $A_2$ appears negatively (because they are satisfied). Next, we delete all literals of $A_1$ and $A_2$ from all formulas in the MLN. The weights of all formulas (which are not deleted) remain unchanged except those formulas in which atoms of $A_1$ or $A_2$ do not share logical variables with other atoms. The weight of each such formula $f$ with weight $w$ is changed to $w \times D(1_{A_1})$ if $A_1$ appears in the clause or to $w \times D(1_{A_2})$ if $A_2$ appears in the clause.

The weight $w(A, i)$ is calculated as follows. Let $F(A_1)$ and $F(A_2)$ denote the set of satisfied formulas (which are deleted) in which $A_1$ and $A_2$ participate in. We introduce some additional notation. Let $V(f)$ denote the set of logical variables in a formula $f$. Given a formula $f$, for each variable $y \in V(f)$, let $i_R(y)$ denote the position of the argument of a predicate $R$ such that $y$ appears at that position in an atom of $R$ in $f$. Then, $w(A, i)$ is given by:

$$w(A, i) = \sum_{k=1}^{2} \sum_{f_j \in F(A_k)} w_j \prod_{y \in V(f_j)} D(i_R(y))$$

We can show that:

**Proposition 3.** *Given an MLN $M$ having singleton atom $A$, the MAP-value of $M$ equals* $\max_{i=0}^{D(1_A)}$ *MAP-value$(M|(A, i)) + w(A, i)$.*

**Example 3.** *Consider a normal MLN $M$ having two weighted formulas $R(x) \vee S(x), w_1$ and $R(y) \vee S(z), w_2$ with domain sizes $D(1_R) = D(1_S) = n$. The MLN $M|(R, i)$ is the MLN having three weighted formulas: $S_2(x_2), w_1$; $S_1(x_1), w_2(n-i)$ and $S_2(x_3), w_2(n-i)$ with domains $D(1_{S_1}) = i$ and $D(1_{S_2}) = n - i$. The weight $w(R, i) = iw_1 + niw_2$.*

**Partial grounding**. In the absence of a decomposer, or when the singleton rule is not applicable, we will have to partially ground a predicate. For this, we heuristically select an argument $i_R$ to ground. Let $M|G(i_R)$ denote the MLN obtained from $M$ as follows. For each argument $i_S \in \text{Unify}(i_R)$, we create $D(i_S)$ new predicates which have all arguments of $S$ except $i_S$. We then update all formulas with the new predicates. For example,

**Example 4.** *Consider a MLN with two formulas: $R(x, y) \vee S(y, z), w_1$ and $S(a, b) \vee T(a, c), w_2$. Let $D(2_R) = 2$. After grounding $2_R$, we get an MLN having four formulas: $R_1(x_1) \vee S_1(z_1), w_1$, $R_2(x_2) \vee S_2(z_2), w_1$, $S_1(b_1) \vee T_1(c_1), w_2$ and $S_2(b_2) \vee T_2(c_2), w_2$.*

Since partial grounding will create many new clauses, we will try to use this operator as sparingly as possible. The following theorem is immediate from [6, 18] and the discussion above.

**Theorem 1.** *PTP-MAP$(M)$ computes the MAP value of $M$.*

## 4 Integer Polynomial Programming formulation for Lifted MAP

PTP-MAP performs an exhaustive search over all possible lifted assignments in order to find the optimal MAP value. It can be very slow without proper pruning, and that is why branch-and-bound algorithms are widely used for many similar optimization tasks. The branch-and-bound algorithm maintains a global best solution found so far, as a lower bound. If the estimated upper bound of a node is not better than the lower bound, the node is pruned and the search continues with other branches. However instead of developing a lifted MAP specific upper bound heuristic to improve Algorithm 1, we propose to encode the lifted search problem as an Integer Polynomial Programming (IPP) problem. This way we can use existing off-the-shelf advanced machinery, which includes pruning techniques, search heuristics, caching, problem decomposition and upper bounding techniques, to solve the IPP.

At a high level, our encoding algorithm runs PTP-MAP schematically, performing all steps in PTP-MAP except the search or conditioning step. Before we present our algorithm, we define schematic MLNs (SMLNs) – a basic structure on which our algorithm operates. SMLNs are normal MLNs with two differences: (1) weights attached to formulas are polynomials instead of constants and (2) Domain sizes of arguments are linear expressions instead of constants.

Algorithm 2 presents our approach to encode lifted MAP problem as an IPP problem. It mirrors Algorithm 1, with only difference being at the lifted conditioning step. Specifically, in lifted conditioning step, instead of going over all possible branches corresponding to all possible counting assignments, the algorithm uses a representative branch which has a variable associated for the corresponding counting assignment. All update steps described in the previous section remain unchanged with the caveat that in $S|(\mathtt{A}, i)$, $i$ is symbolic(an integer variable). At termination, Algorithm 2 yields an IPP. Following theorem is immediate from the correctness of Algorithm 1.

---

**Algorithm 2 SMLN-2-IPP**(SMLN $S$)

**if** $S$ is empty **return** $\langle 0, \emptyset, \emptyset \rangle$
Simplify($S$)
**if** $S$ has disjoint SMLNs **then**
    **for** disjoint SMLNs $S_i...S_k$ in $S$
        $\langle f_i, G_i, X_i \rangle = $ SMLN-2-IPP($S_i$)
    **return** $\langle \sum_{i=1}^{k} f_i, \cup_{i=1}^{k} G_i, \cup_{i=1}^{k} X_i \rangle$
**if** $S$ has a *decomposer* **d** **then**
    **return** SMLN-2-IPP($S|\mathbf{d}$)
**if** $S$ has a *isolated singleton* R **then**
    **return** SMLN-2-IPP($S|\{i_{\mathtt{R}}\}$)
**if** $S$ has a *singleton* atom A **then**
    Introduce an IPP variable '$i$'
    Form a constraint $g$ as '$(0 \leq i \leq D(1_{\mathtt{A}}))$'
    $\langle f, G, X \rangle = $ SMLN-2-IPP($S|(\mathtt{A}, i)$)
    **return** $\langle f + w(\mathtt{A}, i), G \cup \{g\}, X \cup \{i\} \rangle$
Heuristically select an argument $i_{\mathtt{R}}$
**return** SMLN-2-IPP($S|G(i_{\mathtt{R}})$)

---

**Theorem 2.** *Given an MLN $M$ and its associated schematic MLN $S$, the optimum solution to the Integer Polynomial Programming problem returned by SMLN-2-IPP($S$) is the MAP solution of $M$.*

In the next three examples, we show the IPP output by Algorithm 2 on some example MLNs.

**Example 5.** *Consider an MLN having one weighted formula:* $\mathtt{R}(x) \vee \mathtt{S}(x), w_1$ *such that $D(1_{\mathtt{R}}) = D(1_{\mathtt{S}}) = n$. Here, $\mathbf{d} = \{1_{\mathtt{R}}, 1_{\mathtt{S}}\}$ is a decomposer. By applying the decomposer rule, weight of the formula becomes $nw_1$ and domain size is set to 1. After conditioning on R objective function obtained is $nw_1 r$ and the formula changes to $\mathtt{S}(x), nw_1(1-r)$. After conditioning on S, the IPP obtained has objective function $nw_1 r + nw_1(1-r)s$ and two constraints: $0 \leq r \leq 1$ and $0 \leq s \leq 1$.*

**Example 6.** *Consider an MLN having one weighted formula:* $\mathtt{R}(x) \vee \mathtt{S}(y), w_1$ *such that $D(1_{\mathtt{R}}) = n_x$ and $D(1_{\mathtt{S}}) = n_y$. Here R and S are isolated, and therefore by applying the isolated singleton rule weight of the formula becomes $n_x n_y w_1$. This is similar to the previous example; only weight of the formula is different. Therefore, substituting this new weight, IPP output by Algorithm 2 will have objective function $n_x n_y w_1 r + n_x n_y w_1(1-r)s$ and two constraints $0 \leq r \leq 1$ and $0 \leq s \leq 1$.*

**Example 7.** *Consider an MLN having two weighted formulas:* $\mathtt{R}(x) \vee \mathtt{S}(x), w_1$ *and* $\mathtt{R}(z) \vee \mathtt{S}(y), w_2$ *such that $D(1_{\mathtt{R}}) = D(1_{\mathtt{S}}) = n$. On this MLN, the IPP output by Algorithm 2 has the objective function $rw_1 + r^2 w_2 + rw_2(n-r) + s_2 w_1(n-r) + s_2 w_2(n-r)^2 + s_1 w_2(n-r)r$ and constraints $0 \leq r \leq n$, $0 \leq s_1 \leq 1$ and $0 \leq s_2 \leq 1$. The operations that will be applied in order are: lifted conditioning on R creating two new predicates $\mathtt{S}_1$ and $\mathtt{S}_2$; decomposer on $1_{\mathtt{S}_1}$; decomposer on $1_{\mathtt{S}_2}$; and then lifted conditioning on $\mathtt{S}_1$ and $\mathtt{S}_2$ respectively.*

### 4.1 Solving Integer Polynomial Programming Problem

Although we can directly solve the IPP using any off-the-shelf mathematical optimization software, IPP solvers are not as mature as Integer Linear programming(ILP) solvers. Therefore, for efficiency reasons, we propose to convert the IPP to an ILP using the classic method outlined in [25] (we skip the details for lack of space). The method first converts the IPP to a zero-one Polynomial Programming problem and then subsequently linearizes it by adding additional variables and constraints for each higher degree terms. Once the problem is converted to an ILP problem we can use any standard ILP solver to solve it. Next, we state a key property about this conversion in the following theorem.

**Theorem 3.** *The search space for solving the IPP obtained from Algorithm 2 by using the conversion described in [25] is polynomial in the max-range of the variables.*

*Proof.* Let $n$ be number of variables of the IPP problem, where each of the variables has range from 0 to $(d-1)$ (i.e., for each variable $0 \leq v_i \leq d-1$). As we first convert everything to binary, the

zero-one Polynomial Programming problem will have $O(n \log_2 d)$ variables. If the highest degree of a term in the IPP problem is $k$, we will need to introduce $O(\log_2 d^k)$ binary variables (as multiplying $k$ variables, each bounded by $d$, will result in terms bounded by $d^k$) to linearize it. Since search space of an ILP is exponential in number of variables, search space for solving the IPP problem is:

$$O(2^{(n \log_2 d + \log_2 d^k)}) = O(2^{n \log_2 d}) O(2^{k \log_2 d}) = O(d^n) O(d^k) = O(d^{n+k}) \quad \square$$

We conclude this section by summarizing the power of our new approach:

**Theorem 4.** *The search space of the IPP returned by Algorithm 2 is smaller than or equal to the search space of the Integer Linear Program (ILP) obtained using the algorithm proposed in Sarkhel et al. [18], which in turn is smaller than the size of the search space associated with the ground Markov network.*

## 5 Experiments

We used a parallelized ILP solver called Gurobi [9] to solve ILPs generated by our algorithm as well as by other competing algorithms used in our experimental study. We compared performance of our new lifted algorithm (which we call IPP) with four other algorithms from literature: Alchemy (ALY) [11], Tuffy(TUFFY) [14], ground inference based on ILP (ILP), and lifted MAP (LMAP) algorithm of Sarkhel et al. [18]. Alchemy and Tuffy are two state-of-the-art open source software for learning and inference in MLNs. Both of them first ground the MLN and then use an approximate solver, MaxWalkSAT [19] to compute MAP solution. Unlike Alchemy, Tuffy uses clever Database tricks to speed up computation. ILP is obtained by converting MAP problem over ground Markov network to an ILP. LMAP also converts the MAP problem to ILP, however its ILP encoding can be much more compact than ones used by ground inference methods because it processes "non-shared atoms" in a lifted manner (see [18] for details). We used following three MLNs to evaluate our algorithm:

(i) An MLN which we call **Student** that consists of following four formulas,
    Teaches(*teacher,course*) ∧ Takes(*student,course*) → JobOffers(*student,company*);
    Teaches(*teacher,course*); Takes(*student,course*); ¬JobOffers(*student,company*)

(ii) An MLN which we call **Relationship** that consists of following four formulas,
    Loves(*person1 ,person2*) ∧ Friends(*person2, person3*) → Hates(*person1, person3*);
    Loves(*person1, person2*); Friends(*person1, person2*); ¬Hates(*person1, person2*);

(iii) **Citation Information-Extraction** (IE) MLN [11] from the Alchemy web page, consisting of five predicates and fourteen formulas.

To compare performance and scalability, we ran each algorithm on aforementioned MLNs for varying time-bounds and recorded solution quality (i.e., the total weight of false clauses) achieved by each. All our experiments were run on a third generation i7 quad-core machine having 8GB RAM.

For Student MLNs, results are shown in Fig 1(a)-(c). On the MLN having 161K clauses, ILP, LMAP and IPP converge quickly to the optimal answer while TUFFY converges faster than ALY. For the MLN with 812K clauses, LMAP and IPP converge faster than ILP and TUFFY. ALY is unable to handle this large Markov network and runs out of memory. For the MLN with 8.1B clauses, only LMAP and IPP are able to produce a solution with IPP converging much faster than LMAP. On this large MLN, all three ground inference algorithms, ILP, ALY and TUFFY ran out of memory.

Results for Relationship MLNs are shown in Fig 1(d)-(f) and are similar to Student MLNs. On MLNs with 9.2K and 29.7K clauses ILP, LMAP and IPP converge faster than TUFFY and ALY, while TUFFY converges faster than ALY. On the largest MLN having 1M clauses only LMAP, ILP and IPP are able to produce a solution with IPP converging much faster than other two.

For IE MLN results are shown in Fig 1(g)-(i) which show a similar picture with IPP outperforming other algorithms as we increase number of objects in the domain. In fact on the largest IE MLN having 15.6B clauses only IPP is able to output a solution while other approaches ran out of memory.

In summary, as expected, IPP and LMAP, two lifted approaches are more accurate and scalable than three propositional inference approaches: ILP, TUFFY and ALY. IPP not only scales much better but also converges much faster than LMAP, clearly demonstrating the power of our new approach.

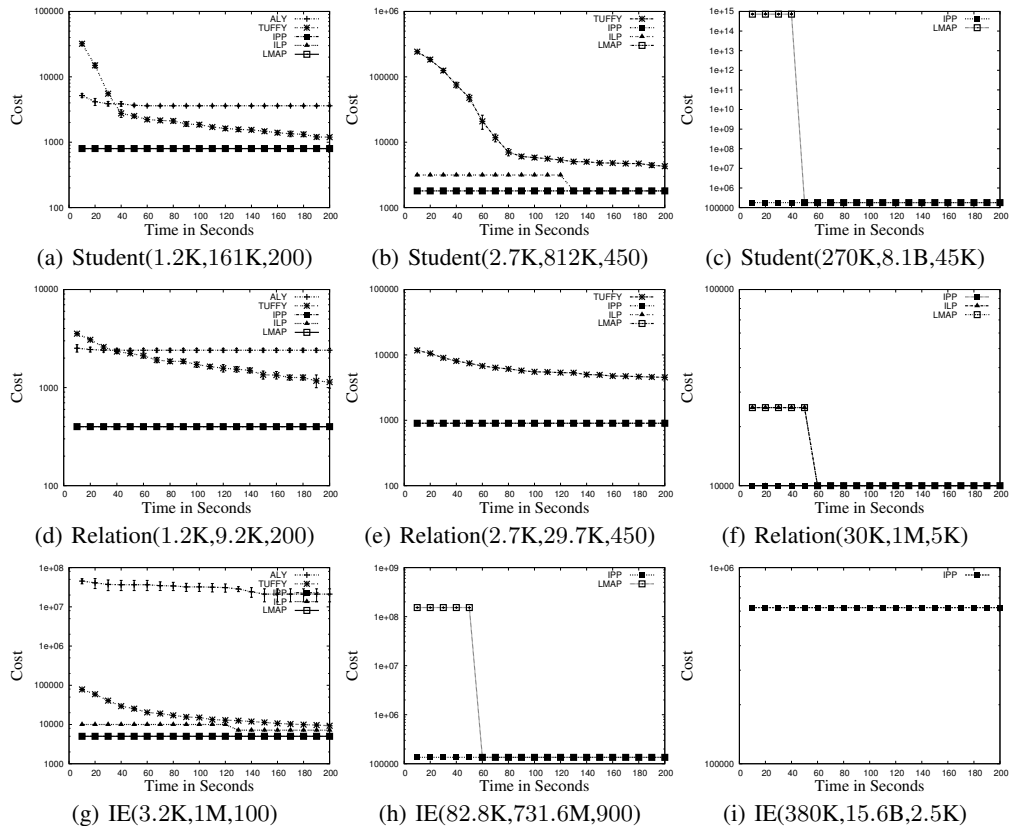

Figure 1: Cost vs Time: Cost of unsatisfied clauses(smaller is better) vs time for different domain sizes. Notation used to label each figure: MLN(numvariables, numclauses, numevidences). Note: three quantities reported are for ground Markov network associated with the MLN. Standard deviation is plotted as error bars.

# 6   Conclusion

In this paper we presented a general approach for lifted MAP inference in Markov logic networks (MLNs). The main idea in our approach is to encode MAP problem as an Integer Polynomial Program (IPP) by schematically applying three lifted inference steps to the MLN: lifted decomposition, lifted conditioning and partial grounding. To solve the IPP, we propose to convert it to an Integer Linear Program (ILP) using the classic method outlined in [25]. The virtue of our approach is that the resulting ILP can be much smaller than the one obtained from ground Markov network. Moreover, our approach subsumes the recently proposed lifted MAP inference approach of Sarkhel et al. [18] and is at least as powerful as probabilistic theorem proving [6]. Perhaps, the key advantage of our approach is that it runs lifted inference as a pre-processing step, reducing the size of the theory and then applies advanced propositional inference algorithms to this theory without any modifications. Thus, we do not have to explicitly lift (and efficiently implement) decades worth of research and advances on propositional inference algorithms, treating them as a black-box.

**Acknowledgments**

This work was supported in part by the AFRL under contract number FA8750-14-C-0021, by the ARO MURI grant W911NF-08-1-0242, and by the DARPA Probabilistic Programming for Advanced Machine Learning Program under AFRL prime contract number FA8750-14-C-0005. Any opinions, findings, conclusions, or recommendations expressed in this paper are those of the authors and do not necessarily reflect the views or official policies, either expressed or implied, of DARPA, AFRL, ARO or the US government.

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
