[Reviews · NeurIPS 2014]

Submitted by Assigned_Reviewer_8

The authors propose a new method of performing lifted inference on Markov Logic Networks. The essence of the idea is to encode the MLN as an integer polynomial program, which is then transformed into an integer linear program (which could be solved with a conventional solver such as Gurobi or CPlex).

The lifting as preprocessing idea due to Sarkhel et al. is appealing, and this paper extends it using ideas from probabilistic theorem proving. The idea is to extend the list of symmetries recognized by this lifted inference approach (Algorithm 1). Indeed the experiments bare out that this approach improves over the previously stated algorithms. I believe this result will be interesting to people who work on lifted inference of MLNs and perhaps more broadly.

The experimental validation could be improved by considering more sophisticated MLNs (say with features that depend on the
data or encoding of standard logistic regression). The most long-standing criticism of the lifted approaches are that they exploit symmetries that are not often found in real data (at a level which is unreasonable for "real world data"). If the authors believe that their method works on such datasets that would strengthen the claims of the paper.

The paper is well prepared and easy to follow.

Summary: Well prepared paper that advances our understanding of lifted inference for MLNs.

Submitted by Assigned_Reviewer_15

This paper proposes to use lifted inference rules to generate an IPP problem for solving MLN MAP tasks.

The paper is very well written. Sections 2 and 3 give a nice tutorial on lifted MAP inference.
The only thing I could not figure out is what PTP-MAP does when you have relations with 0 arguments, say, a classical MRF. Then, none of the if statements hold, and the last return has nothing to partially ground?

Could there be a mistake in Theorem 3, Equation 2? I think that O(2^{log(d)^k}) is a quasi-polynomial complexity, and it is strictly harder than O(d^n). So the search space is not polynomial in d, only quasi-polynomial.

The assumption that the MLN has no self-joins is strong, and limits the usefulness of the algorithm.

There seems to be a connection between schematic MLNs and first-order d-DNNF circuits (domain size expressions become domain variables, etc.).

The paper would be much stronger if it had experiments on real-world data, and if it would compare to RockIt.

Lines 205 has a typo. Line 208 should say 'after'.

Summary: I enjoyed reading this paper, and it presents a neat idea. I have only two concerns: the contribution is rather lightweight (only page 6 contains new material, everything else is background), and I believe the main Theorem 3 has an error.

Submitted by Assigned_Reviewer_32

Paper Summary: The paper presents an approach to encode the MAP inference problem in Markov logic networks as an integer polynomial program (IPP) using three steps: lifted decomposition, lifted conditioning, and partial grounding. In order to solve the IPP, it converts it into an integer linear program using [25], and then resorts to using the standard Gurobi solver.

Originality: The techniques of lifted decomposition, lifted condition and partial grounding have previously been used for lifted MAP inference for MLNs previously [6,18]. So the main contribution of the paper is to realise that the MAP inference problem can be formulated as an IPP. Solving the problem in this way does seem to result in faster inference, but the practical usefulness of such an approach is unclear (see below).

Clarity: The paper is a difficult read. The main problem is that the paper is not self-contained, and relies on the reader's familiarity with [6]. It also uses several terms without definition, for example,
- Line 162: What are the 'arguments' i_R and j_S?
- Line 172: What are the 'atoms of a predicate'? This is confusing since the previous section describes atoms as predicates (line 94).
- Line 179: What is the function D?
While these terms become clearer from context, especially through the examples provided, they still hurt the paper's clarity. The general NIPS audience will not be familiar with MLNs, which have been proposed relatively recently. While I understand that it is difficult to define everything concretely within the page limit, it would help to publish an accompanying self-contained technical report.

Significance: The experiments compare to the state of the art [11,14,18], and demonstrate significant speed-up for three MLNs. However, some details are missing. How were the weights for the MLN estimated? Is there a separate training set? While the performance of the methods has been reported only in terms of the cost, can it also be reported in terms of the accuracy of the output? Although the main aim of the paper is better optimisation, it's case would be made much stronger if better optimisation leads to better accuracy. Also, is there any theoretical bound on the size of the IPP obtained using the proposed method? If not, at least the empirical size of the IPP and the ILP should be reported.

Post-rebuttal comments: My main concern with the paper is that the experiments are conducted on artificial MLNs (with hand-tuned weights), and the performance is reported purely in terms of the cost and not the accuracy. Nonetheless, since the main objective of the paper is improved inference and the experiments are able to demonstrate the benefit of the proposed method in this regard, I vote for accepting the paper. If accepted, please include the explanations provided in the rebuttal, as well as the size of the IPP and ILP. It would also be beneficial to release a longer, self-contained tech report instead of relying on the readers familiarity with the state of the art.
Summary: An interesting, but difficult to read paper. Experiments show improvements over the state of the art, but some experimental setup details are missing.
Author Feedback
Author rebuttal: Thanks to all the reviewers for their helpful comments.

Assigned_Reviewer_15

Handling atoms with zero arguments/propositions in MRFs: We can associate a predicate having one argument with each random variable in the (binary) MRF and set the domain size of the argument to “1”. For example, (R v S) can be written as R(x) v S(y) where \Delta x = \Delta y = {0}. Notice that the binomial rule can be applied to such atoms. Also note that normal forms explicitly do this. (We can easily come up with cleverer encodings in which the domain size need not be 1 and the number of predicates need not be equal to the number of propositions in the MRF. However, this is outside the scope of this paper. Our algorithm is applicable to such cases.)

Regarding Theorem 3:
The statement of the theorem is correct but there is a mistake in the proof. On line 323 we said that we need to introduce O((log_2 d)^k) variables. Instead we only need to introduce only O(log_2 d^k) variables (i.e.- O(k log_2 d) variables). For example, suppose that we have a multi-linear constraint “ab-a-b<=1” where 0 ≤ a ≤ d and 0 ≤ b ≤ d. We can convert this to 0-1 integer constraint by introducing log_2d (bits) variables to model “a”, log_2d variables to model “b” and log_2(d^2) bits to model “a.b”. (since a.b is bounded by d^2). We will change the proof to reflect this. Moreover, the last line should be O(d^{n+c}) where “c” is a constant (we are assuming that the number of formulas and the number of multi-linear and non-linear terms is bounded by a constant).

We will also fix the typos you mentioned in the camera-ready version.

Regarding Self-Join:
We have assumed that there are no self-joins to simplify our presentation. However our approach works even in the presence of self-joins and one of the MLNs used in our experiment (the Information Extraction MLN) has self-joins. The two algorithms mentioned in the paper remain unchanged for self-joins. However, the computation of weights will be different and the technical details are quite complicated.

A key point to note is that the general approach of “schematically running PTP or other search-based algorithms such as the ones proposed by Broeck et al. and Poole et al. to obtain an IPP” will remain the same.

Regarding our contribution in Section 3:
We could easily rewrite the paper without relating it to previous work and directly stating the encoding details (e.g., we can have section 3 on how to encode a decomposer; section 4 on how to encode the binomial rule; section 5 on how to encode MAP-specific inference rules; etc. ). However, we prefer simplicity and we took some great care in making sure that the method is not only extensible (notice that Algorithm 1 sketches a framework to plug-in any newly discovered lifting rule) but also easy to implement.

Regarding experiments:
Please see our response to Assigned_Reviewer_8

Assigned_Reviewer_32

In line 162 i_R and j_S represents respectively i-th argument of R and j-th argument of S. The argument notation is mentioned at line 130. e.g. - In the predicate R(x, y) the term 'x' is denoted by 1_R (i.e.- first argument of R). Similarly in the line 179 the function D(x), which is mentioned at line 131, is the domain size of an argument x. e.g. - for the predicate S(x) with domain of x being {A, B} we say that D(1_S) = 2.

- Line 172: atoms of a predicate “R” is atoms in formulas having predicate symbol “R”

For the experiments we did not learn the weights of the MLNs (for many large problems used in our study using existing weight learning methods (in Alchemy) is not feasible). Therefore, we set them to reasonable values (we will make these MLNs available). Since our goal is faster and better inference we reported cost vs time. The cost is the total weight of unsatisfied clauses, which is inversely related to the MAP value. Thus, smaller the cost the better the accuracy.

Regarding the size of the IPP/ ILP:
The size of the IPP and ILP depends on the structure of the problem. In the best case scenario the size of the IPP (i.e.- no of variables in the IPP) is equal to the number of predicates in the MLN, however in the worst case (the scenario where MLN is non-liftable) the size of the IPP is equal to the total no of ground atoms. Due to space constraints, we were not able to mention these sizes explicitly however they follow directly from Algorithm 2 since we introduce an IPP variable for each conditioned atom. Theorem 3 (informally) mentions the size of the ILP in terms of the size of the IPP. For the three MLNs mentioned in our experiments section we will report the size of the IPP as well as the ILP in the camera-ready version.

Assigned_Reviewer_8

We are well aware of the limitations of any lifted approach. For some MLNs (which we will refer to as non-liftable MLNs) we will not be able to exploit symmetries. However, even in such cases our approach will be no worse than ground inference (Theorem 4). In fact the main motivation of our approach (i.e.- to encode the problem as an IPP instead of implementing a specific branch-and-bound solver) is to use decades of advances in propositional inference. In our experiments, one of the three MLNs, (the Citation Information Extraction) has been taken off the shelf from the Alchemy website and our experiments show that our approach performs very well on that MLN.